# Integration of palliative care into phase I oncology trials: A qualitative interview study with patients, informal caregivers, and healthcare providers

Frederick Daenen [1,2]*, Anne van Driessche[1,2], Anne-Lore Scherrens[1,3], Annelies Janssens[4], Sylvie Rottey[5], Lore Decoster[6], Peter A. J. Stevens[7], Koen Pardon[1,2], Aline De Vleminck[1,2], Kim Beernaert[1,3]

1 End-of-life Care Research Group, Vrije Universiteit Brussel (VUB) & Ghent University, Brussels, Belgium, 2 Department of Family Medicine and Chronic Care, Vrije Universiteit Brussel (VUB), Brussels, Belgium, 3 Department of Public Health and Primary Care, Ghent University, Gent, Belgium, 4 Department of Pulmonology and Thoracic Oncology, Antwerp University Hospital, Antwerp, Belgium, 5 Department of Medical Oncology, Ghent University Hospital, Ghent, Belgium, 6 Department of Medical Oncology, Oncologisch Centrum, Universitair Ziekenhuis Brussel, Vrije Universiteit Brussel, Brussels, Belgium, 7 Department of Sociology, Faculty of Political and Social Sciences, Ghent University, Ghent, Belgium

* Frederick.daenen@vub.be

## Abstract

### Background

Patients with advanced cancer participating in phase I clinical trials often face limited survival while experiencing significant symptom burden. Despite evidence supporting early palliative care integration alongside active cancer treatment to improve quality of life, the role of palliative care in phase I trials remains unclear.

### Objective

To explore perspectives of patients, informal caregivers, and healthcare providers on quality of life and palliative care in phase I oncology trials, including perceived benefits, barriers, and integration strategies.

### Methods

We conducted a multi-perspective qualitative interview study across three Belgian university hospitals from September 2022 to July 2024, using convenience and snowball sampling. The semi-structured interviews were analyzed using qualitative content analysis. Ethical approval was gained from the relevant institutions.

### Results

Participants included sixteen patients, five informal caregivers, twelve phase I staff, six oncologists, five palliative care specialists, and four general practitioners. Patients

**Data availability statement:** The data underlying this study consist of qualitative interview transcripts containing potentially identifying and sensitive information from patients, family caregivers, and healthcare professionals. In accordance with the approval conditions of the Medical Ethics Committee of UZ Brussel/VUB, these data cannot be shared publicly to protect participant confidentiality. Pseudonymized excerpts relevant to the findings can be shared upon reasonable request. Data requests may be sent to either the first author or the Medical Ethics Committee of UZ Brussel/VUB (Ethiek@uzbrussel.be). Any data sharing will only occur after formal approval from the Ethics Committee and in accordance with applicable data protection regulations.

**Funding:** Project funded by Kom op tegen Kanker (Stand up to Cancer), the Flemish cancer society (13719) (URL: https://www.komoptegenkanker.be/). This grant was awarded to author KB. The funder had no role in study design, data collection and analysis, decision to publish, or preparation of the manuscript.

**Competing interests:** The authors have declared that no competing interests exist.

generally reported positive experiences with trial participation, often viewing it as a final opportunity that provided hope and structure. However, quality of life support was inconsistently addressed and largely reactive. While patients reported feeling supported, non-trial providers and caregivers noted limited person-centered care. No systematic approach for introducing palliative care was in place. Palliative care was rarely discussed, hindered by misconceptions such as equating palliative care with terminal care, reluctance from patients and clinicians, and lack of communication between providers. Participants suggested the introduction of routine yet flexible palliative care conversations, as well as improved communication between providers, as strategies towards integration.

## Conclusion

Despite recognized care needs, palliative care is not systematically integrated in phase I oncological trials. Quality of life remains a secondary concern. Integrating palliative care in a structured yet flexible manner could support more holistic, patient-centered care. These findings underscore the need to normalize palliative care as a complementary component of phase I oncological trials.

## Introduction

Patients with advanced cancer who have exhausted standard treatment options can be offered participation in phase I clinical trials [1]. These trials primarily focus on evaluating the toxicity and appropriate dosing of novel therapies, unlike phase II and phase III trials which assess therapeutic efficacy [2]. Patients are motivated to participate in these trials out of hope for therapeutic benefit, with recent innovations in early-phase drug development having led to improved response rates and prolonged tumor response durations [3]. Yet, phase I trial participants face a limited survival outlook, with the majority passing away within a year [4,5]. Importantly, while eligibility criteria for phase I trials generally require patients to have a relatively good performance status (ECOG 0–1) [6], retrospective studies show that these patients experience a high symptom burden, comparable to or even greater than that of patients who are not enrolled in clinical trials [6–8].

Despite patients' significant care needs, the primary focus in phase I studies remains on experimental treatment [6]. Patients are often confronted with a choice between pursuing experimental therapy or transitioning fully to comfort-oriented, palliative care [9,10]. In this context, healthcare providers play a crucial role not only in screening and enrolling eligible patients, but also in providing clear information about the trial's aims, limitations, and potentially also about palliative care, balancing transparency with protection from potential overtreatment [11].

Palliative care is a holistic approach that addresses physical, psychological, social, and spiritual needs, and may be provided by palliative care specialists or integrated through basic palliative care principles by other healthcare providers [12]. This may

include a holistic focus on the quality of life of patients in the broadest sense, as well as more specific actions such as the discussion of advance care planning and palliative care services. Early integration of palliative care alongside active cancer treatment has been shown to improve quality of life for patients and their families [13–16]. For patients who are no longer candidates for curative treatment, early discussions about goals of care, advance care planning, and the potential initiation of palliative services are critical [13,17,18]. In Belgium, phase I trials are primarily conducted at large academic centers, with other hospitals referring patients to these centers. Coordination of care can become more complex when patients transition between centers or departments, making attention to quality of life, advance care planning, and palliative care even more essential.

Palliative care in Belgian hospitals can be provided either in a specialized way, provided in palliative care units by a team including specific palliative care experience, or in a generalist way, by multidisciplinary teams supporting patients not in a palliative care unit. At present, little is known about whether palliative care is integrated in phase I trials: whether and how palliative care is discussed with this patient population, when it is initiated, and how patients and families experience their quality of life during participation in experimental studies. Given the significant care needs and limited survival outlook of trial participants, early integration of palliative care could offer substantial benefits such as improving the quality of life of patients and their families.

### Objectives

The goal of this project is to gain insights into the following three research questions, as perceived by trial participants with cancer, informal caregivers, and various types of healthcare providers (phase I staff, referring oncologists, general practitioners (GPs), and palliative care specialists). As experiences and expectations around palliative care may differ markedly between patients, informal caregivers, and the various types of healthcare providers throughout patients' care trajectories, exploring these multiple perspectives allows for a more comprehensive understanding of current practices and potential integration.

1. How are aspects of patients' quality of life recognized, monitored, and supported within phase I oncological trials?

2.

 a. How is palliative care currently addressed in phase I oncological trials?

 b. What are patients' attitudes towards palliative care provision?

3. What are the perceived benefits, barriers, and strategies to integrating palliative care into phase I oncological trials?

## Method

### Study design

This study employed a qualitative descriptive research design using semi-structured face-to-face interviews [19]. A qualitative descriptive design was chosen as it allows for the in-depth studying of unexplored topics, with semi-structured interviews allowing flexibility for the uncovering of potentially unforeseen topics related to the research questions. We applied a multi-perspective essentialist approach, prioritizing a broader data collection of multiple types of participants over a more idiosyncratic hermeneutical interpretative approach [20]. This study was conducted and reported following the Consolidated Criteria for Reporting Qualitative Research (COREQ) guidelines (S1 File COREQ checklist) [21].

### Population and setting

To comprehensively explore perspectives on integrating palliative care into phase I clinical trials, we included cancer patients, informal caregivers, and healthcare providers. The latter group comprised phase I staff (physicians, nurses,

study coordinators), referring oncologists, palliative care specialists (i.e., members of the hospital palliative support team), as well as GPs as they may play a pivotal role in the organization of palliative care services and advance care planning [22]. We targeted at least four participants per category and at least five healthcare providers per site. The study was conducted at three Belgian university hospitals: Ghent University Hospital and Antwerp University Hospital (both with dedicated oncological trial units) and UZ Brussel (no dedicated unit but referring patients to such units and hosting small-scale trials).

Inclusion criteria required participants to be Dutch-speaking and aged 18 or older. Patients had to be enrolled in any phase I oncological trial; informal caregivers were those caring for a patient in such a trial. Healthcare providers needed at least one year of experience in their current role. Non-phase I staff needed experience with phase I patients, either through referrals or managing patients in trials. Providers were not necessarily treating the patients interviewed.

### Recruitment

Participants were recruited using convenience and snowball sampling. Recruitment started at Ghent University Hospital on September 1st 2022, with Antwerp University Hospital and UZ Brussel added as study sites on March 1st 2024. Recruitment ended at all sites on July 2nd 2024. Patients and informal caregivers were recruited from the dedicated oncological trial units at Ghent University Hospital and Antwerp University Hospital. Study coordinators introduced the study to patients and, when present, informal caregivers, and forwarded the contact details of interested individuals to the research team, who contacted them by telephone. Non-participation was mainly due to health issues or reluctance to engage in further research. Healthcare professionals were recruited via networks and snowballing.

### Data collection

Interviews took place between September 19th 2022 and July 2nd 2024 at locations chosen by participants in a quiet space. Patients and caregivers were interviewed during or after hospital visits, and healthcare providers during working hours. In two cases, patients and informal caregivers were interviewed together. A stakeholder-informed topic guide was tailored for each group, covering expectations toward the trial, quality of life, advance care planning, and palliative care (S2 File Topic guides). To minimize bias, participants first shared their understanding of palliative care before being provided with a broad, holistic definition. Sociodemographic data were collected via brief questionnaires.

Interviews were conducted in Dutch by two postdoctoral researchers (FD, AVD), one PhD student in psychology, and two trained master's students in healthcare. Interviewers took post-interview field notes. All interviews were audio-recorded, transcribed verbatim, and ranged from 28–117 minutes (patients/caregivers) and 32–95 minutes (healthcare providers). No transcripts had to be returned to participants. The research team met regularly to discuss findings, adapt the topic guide, and assess data saturation, which was reached when no new key themes (e.g., new strategies or barriers) emerged.

### Ethical considerations

This study was approved by the central ethics committees of Ghent University Hospital (6702022000253) and UZ Brussel (1432023000251) and by the local committee of Antwerp University Hospital. All participants received study information and gave written informed consent. Participants were informed they could withdraw at any moment and were also asked whether they agreed to be potentially contacted in the future for member checking.

### Data analysis

Data were analyzed using qualitative content analysis with NVivo 14 [23]. Qualitative content analysis was selected for its suitability in systematically identifying categories within large datasets and supporting descriptive exploration

in understudied areas [20,24,25], combining inductive and deductive approaches as described below [26,27]. As our semi-structured interviews do not allow direct comparison of category prevalence, we focus on a narrative presentation of findings without quantification [20].

The coding process involved two coders (FD, AVD) first familiarizing themselves with the data, then independently performing 'open coding' on a subset of five interviews, selected according to the diversity principle, i.e., having at least one interview per participant category and hospital. Open coding refers to inductively attributing codes to small, meaningful parts of the transcripts relating to our research questions, with codes describing the essence of the coded text while staying close to the raw data [26,28]. The coders then collaboratively and iteratively developed a coding tree, grouping inductively applied open codes into (sub)categories, deductively informed by categories of interest for the research questions: 1) quality of life, 2) current palliative care, and 3) potential integration of palliative care, broadly including palliative care communication, advance care planning, and focusing on quality of life. In a next step, the coders developed a hierarchical coding scheme based on the coding tree, including definitions and applicability rules for all (sub)categories. After the development of the initial coding scheme, trial coding was conducted by both coders independently applying the coding scheme to another subset of selected interviews until a strong interrater reliability of.70 was reached, assessed with Cohen's kappa, iteratively refining the coding scheme after each round of trial coding. This threshold was reached after the independent coding of four interviews, resulting in a finalized coding scheme (see Table 1) [29,30]. The finalized coding scheme was then applied to all transcripts, with coders dividing the remaining transcripts between them. After coding, key findings within each subcategory were synthesized per participant category using a table with each cell representing the main findings per theme (columns) and interview (rows). Representative quotes were selected to illustrate the key findings in each subcategory. To enhance the validity of the findings, we applied member checking, sharing a synthesis of

**Table 1.** Coding scheme.

| Category | Description | Subcategories & description | Applicability | Example codes |
|---|---|---|---|---|
| Quality of life in phase I studies | This category includes subcategories related to the quality of life of participants in Phase I studies and their relatives, as experienced throughout the Phase I trajectory. | Subcategories are classified based on:<br>1. The reasons patients & relatives participate in the study (what do they hope to achieve?);<br>2. The experienced quality of life, in the broadest sense, of the patient and their relative during the study;<br>3. The extent to which study personnel do or do not take their quality of life into account (separately from the concept of palliative care). | Statements are coded regardless of whether they come from the patient, relative, or healthcare provider, but they must concern the quality of life of the patient or their relative. | Participating due to a final life wish<br>Being able to live as normally as possible<br>Feeling lonely in the hospital room |
| Palliative care in the context of phase I studies | This category includes all subcategories related to palliative care in the phase I trajectory. Palliative care is considered holistically but must be explicitly understood or discussed in terms of palliative care – general quality-of-life considerations are not coded under this category. | Subcategories are classified based on:<br>1. Patients' & relatives' attitudes towards palliative care, specifically also regarding advance care planning;<br>2. How palliative care is or is not discussed with the patient by healthcare providers, and in what manner;<br>3. Communication about palliative care for the patient between different healthcare providers. | Statements are coded as long as they explicitly relate to the concept of palliative care within the study trajectory, starting from the referral. | Discussing palliative care is for later<br>Palliative care is not discussed with the patient<br>Palliative care is not mentioned at the time of referral |
| Integration | This category includes all reflections from patients, relatives, or healthcare providers on the potential integration of palliative care within the study trajectory. | Subcategories are classified based on the characteristics of the integration:<br>1. The benefits of such integration;<br>2. The concrete approach to such integration, including role distribution;<br>3. The barriers to such integration. | Disadvantages are placed under barriers. Barriers are also interpreted as reasons why palliative care is not yet provided. However, the absence of palliative care is coded in the above category (discussion of palliative care). | Patients need to know the reality of their situation<br>The oncologist should mention palliative care at the time of referral<br>It is too early for this population |

the results with a subset of 8 participants, including representatives from each participant type, to verify the interpretation and add further nuance where needed. Interviewed participants supported the shared findings – small nuances to the findings emerging from the member checking interviews are integrated in the results section below.

## Results

A total of 48 participants were interviewed: sixteen patients, five informal caregivers, twelve phase I staff, six oncologists, five palliative care specialists, and four GPs. Participants characteristics are displayed in Table 2. Cancer types and stages varied across participating patients. Phase I staff members held a diverse range of roles, including four physician-investigators, two heads of department, three nurses, and three study coordinators.

The findings are grouped according to the three research aims, each including three subcategories: first, quality of life, including reasons for participation, experienced quality of life, and staff efforts to support quality of life; second, palliative care, including patient attitudes, current communication with patients, and current communication between healthcare providers; third, the potential integration of palliative care in the future, including benefits, barriers, and strategies. For each subcategory, we narrate the key findings and include supporting representative quotes. A summary of the (sub)categories can be found in Fig 1.

**Table 2. Participant characteristics.**

| | | Patients | Informal caregivers | Phase I staff | Oncologists | Palliative care specialists | General practitioners |
|---|---|---|---|---|---|---|---|
| **Total N** | | **16** | **5** | **12[a]** | **6** | **5** | **4** |
| Sex | Female | 4 (25%) | 2 (40%) | 8 (67%) | 6 (100%) | 4 (80%) | 3 (75%) |
| | Male | 12 (75%) | 3 (60%) | 4 (33%) | 0 (0%) | 1 (20%) | 1 (25%) |
| Age | 18-30 | 0 (0%) | 1 (20%) | 2 (17%) | 0 (0%) | 0 (0%) | 0 (0%) |
| | 31-50 | 1 (6%) | 0 (0%) | 8 (67%) | 5 (83%) | 1 (20%) | 1 (25%) |
| | 51-70 | 10 (63%) | 3 (60%) | 2 (17%) | 1 (17%) | 4 (80%) | 3 (75%) |
| | 70+ | 4 (25%) | 1 (20%) | 0 (0%) | 0 (0%) | 0 (0%) | 0 (0%) |
| | missing | 1 (6%) | 0 (0%) | 0 (0%) | 0 (0%) | 0 (0%) | 0 (0%) |
| Location | UZ Brussel | 0 (0%) | 0 (0%) | 2 (17%) | 2 (33%) | 0 (0%) | 0 (0%) |
| | Ghent University Hospital | 12 (75%) | 4 (80%) | 6 (50%) | 2 (33%) | 2 (40%) | 0 (0%) |
| | Antwerp University Hospital | 4 (25%) | 1 (20%) | 4 (33%) | 2 (33%) | 2 (40%) | 0 (0%) |
| | Outside hospitals | 0 (0%) | 0 (0%) | 0 (0%) | 0 (0%) | 1 (20%) | 4 (100%) |
| Education | Higher education | 10 (63%) | 3 (60%) | | | | |
| | High school | 4 (25%) | 1 (20%) | | | | |
| | None | 1 (6%) | 0 (0%) | | | | |
| | missing | 1 (6%) | 1 (20%) | | | | |
| Marital status | Married | 14 (88%) | 5 (100%) | | | | |
| | Divorced | 1 (6%) | 0 (0%) | | | | |
| | Widowed | 1 (6%) | 0 (0%) | | | | |
| Religion | Catholic | 6 (38%) | 1 (20%) | | | | |
| | Atheist | 7 (44%) | 2 (40%) | | | | |
| | Other | 3 (19%) | 1 (20%) | | | | |
| | missing | 0 (0%) | 1 (20%) | | | | |
| Years of experience in current role (M, SD)[b] | | | | 8.6 (2.3) | 11.3 (2.8) | 15.3 (3.8) | 23.3 (5.1) |

a. Consisting of: 6 physicians, 3 study coordinators, 3 nurses.

b. Means (M) and standard deviations (SD) are displayed for years of experience of healthcare professionals in their current role.

| Quality of life in phase I clinical trials | Current palliative care | Potential integration of palliative care |
|---|---|---|
| **Motivations for participation:** Hope, symptom relief, and control drive participation despite limited expectations. | **Patient attitudes toward palliative care:** Palliative care is equated with terminal care, leading to resistance. | **Perceived benefits:** Normalizing palliative care for both patients and providers, enhancing holistic, patient-centered care. |
| **Experienced quality of life:** Patients report positive experiences, though burdens and uncertainty remain. | **Communication with patients:** Discussions are rare and often avoided to preserve hope. | **Barriers:** Misconceptions, unclear roles, and reluctance to take away hope impede integration. |
| **Staff support:** Support remains reactive rather than proactive, with a focus on physical rather than holistic care. | **Communication between providers:** Fragmented communication hinders coordinated palliative care. | **Strategies:** Routine, flexible discussions and improved communication. |

**Fig 1. Summary of (sub)categories and findings.**

### Quality of life in phase I clinical trials

**Quality of life as a motivation for participation.** Participants reported multiple reasons for enrolling in a phase I clinical trial, many of which reflected their hopes of preserving or improving quality of life. Hope for therapeutic benefit was the most typical factor, particularly among younger patients – even though they acknowledged that physicians had been transparent about the lack of guaranteed improvement. This is illustrated by the following quotes from a patient (P) and their informal caregiver (IC):

*"(P) I hope – expectations are something else – I hope it works, yes.*

*(IC) While they haven't... given her any reason for hope, that's not it. That has been made clear, it's an experiment. So, it doesn't cure, but it can of course prolong and keep (her) in good condition.*

*(P) So nothing about recovery or so, I know that's not possible."* (Patient & informal caregiver)

Many participants saw trial participation as their last available option for treatment – a choice that could stabilize their condition and provide structured follow-up, reducing fears of being left without support and maintaining quality of life in the face of uncertainty. Some also cited non-personal motivations, such as altruism and contributing to science: *"Many say,*

you help science, and potentially other people." *(Phase I study coordinator)* Others framed their decision relationally, seeking to extend life for loved ones or reach important milestones. Healthcare providers largely echoed these motivations, noting that patients who sign up often have a strong wish for continued treatment. However, some providers pointed out that patients may feel pressured to participate when suggested by their oncologist.

**Experienced quality of life and well-being.** Patients overwhelmingly described their quality of life and well-being during the trial as positive, specifically due to symptom improvement and to the psychological boost and hope provided by having this, as they often perceived it, 'final chance'. This is illustrated particularly well by the following quote where a patient notes that participation grants perspective that otherwise would not have been there: *"[…] the study gives hope, of course […]. If you have nothing left, I think that's more difficult." (Patient)*

Patients often expressed great gratitude for being afforded the opportunity to participate in the trial. They also noted feeling secure due to the 24/7 availability of phase I staff, signifying stronger monitoring than during standard oncological treatment: *"You feel safer. They emphasize that too. When they say, 'you can reach me tonight if necessary', it gives a kind of comfort. That's important too." (Patient)*

Nevertheless, challenges were also reported. Besides the potential side effects, which varied greatly across treatments, uncertainty regarding treatment effectiveness and long-term outcomes was a source of distress for some, with some patients describing a sense of being 'guinea pigs': *"I am a guinea pig. I'm not really a patient here, if you know what I mean – they're observing what the drug does to me in a strictly scientific way." (Patient)*

Additionally, the time demands of the trial, including frequent hospital visits and extensive testing, were perceived as burdensome by all types of interviewed participants, although less so by the patients than the other interviewees.

**Staff efforts to support quality of life.** Phase I staff expressed routinely verbally inquiring about quality of life, which patients echoed, who generally indicated feeling heard. Phase I staff noted that they would refer patients to psychologists or dieticians, although only reactively upon noticing a clear need, rather than proactively. However, informal caregivers and non-trial healthcare providers were less positive, highlighting a gap in psychosocial support. Notably, while phase I staff indicated that they understood quality of life to be highly subjective based on each patient and what they deemed to be important for their own lives, this was not reflected by the perception of informal caregivers and non-trial providers, who indicated that inquiring into patients' quality of life remains surface-level and that how the patient is 'really' doing is often not addressed:

*"It's never about quality of life. It's always about objective things [medical results]." (GP)*

*"What the patient reports, often those are questionnaires or an interview with a psychologist, but that's more in phase II or phase III studies. What the patient thinks is less considered in phase I." (Oncologist)*

While phase I staff indicated that patients sometimes complete quality of life questionnaires, this was inconsistent across studies. When such questionnaires were administered, staff members admitted that the results were typically not available to them, only being collected for study purposes and not being acted upon. While personal goals and wishes of patients were not always explicitly discussed (e.g., preferences for timing of treatments), phase I staff did report making efforts to balance patients' personal preferences with trial protocol requirements, although this also highlights the concessions that patients may have to make to remain with the boundaries of the study protocols:

*"That's not always in our hands, we're somewhat bound to the protocols that must be followed. But of course, the patient's wish is always the most important. We certainly take that into account. If they report something like, look, that really wouldn't work, then we try to adjust." (Phase I study coordinator)*

 

### Current palliative care in phase I clinical trials

**Patient attitudes towards palliative care.** Patients mostly held negative attitudes and resistance towards the concept of palliative care, associating it with terminal care. Many did not consider themselves as 'palliative' – nor terminal – and perceived conversations about palliative care as premature or unnecessary, as illustrated by the following quotes:

*"I've never been in that mindset of 'it won't be long – I have two months to live.' I've never been there. It's also never been said to me. If they told me tomorrow, 'sorry, you have two months left.' Only then would I consider myself truly palliative and prepare my mind for death. But I'm not occupied with that yet. That's not the case now, not yet." (Patient)*

*"(And has the word palliative care ever come up? Has the doctor or a nurse ever mentioned it?) No. But I think they also know not to bring it up with me." (Patient)*

Most patients interviewed appeared unaware of the broader scope of palliative care and benefits it could offer; those who were aware, attributed their knowledge to doing their own research, rather than being informed by healthcare providers.

**Communication about palliative care with patients.** While phase I staff acknowledged the importance of – and their responsibility in – discussing palliative care, there was no systematic integration of palliative care conversations within the phase I trial setting. Conversations about palliative care were rarely held in the context of the phase I trial setting, either remaining unaddressed or being postponed until patients' condition worsened significantly, in which case staff would refer to palliative care specialists who would thus only be involved reactively in their care trajectories. They also noted that the study population has a particularly very strong will to live, indicating to them that discussing palliative care is likely not welcomed. Some phase I staff felt that the context of a phase I trial did not easily allow for a focus on palliative care:

*"Yes, I think, time-wise. We have to discuss a lot of practical things, a lot about the study. Hm, and it also falls some-where within that… but it's also about balancing giving hope while we are focused on the side effects, right? Yes, sometimes that just falls by the wayside, I must say." (Phase I physician)*

Taking into account this difficult context, phase I staff often indicated that the task of bringing up palliative care may be more suited for the referring oncologist, as patients typically do not remain sufficiently long in the trial to build up a strong connection:

*"In phase I, [palliative care] is not immediately actively started, I think, or only rarely. Because those patients are usually referred by the treating oncologist, since you're somewhat the interim oncologist. They've usually been in a palliative trajectory for quite some time. So I'm not with the patient... That might be something I pay too little attention to. But with the idea that, yes, that might already have been started by the treating oncologist." (Phase I physician)*

However, referring oncologists noted that referral to a phase I trial was a barrier for them to start such conversations, as it seemed too much of a mixed message to the patient to them. Importantly, some healthcare providers, whether phase I staff or referring oncologists, expressed that they did bring up palliative care with their patients, but attributed this to a per-sonal conviction of its benefits rather than a systematic approach. When palliative care was addressed, many healthcare providers avoided using the term 'palliative', instead opting for more neutral language such as comfort care. However, some providers, mainly palliative care specialists, expressed the importance of using the term 'palliative' as to avoid the patient not understanding the severity of the situation, while offering an explanation to the patient of what the term precisely entails.

**Communication about palliative care between healthcare providers.** Information exchange about palliative care between healthcare providers was inconsistent. While the referring oncologist is asked to provide a large amount of information to the phase I unit upon their referral, details on palliative care status, needs, or conversations are neither asked, nor systematically shared. While some oncologists chose to provide this information organically, this was limited and based on personal conviction rather than standardized practice. As such, phase I staff also often were unaware of whether palliative care conversations had been held with the patients prior to their enrollment in the trial trajectory, or even whether palliative care support at home had been set up:

*"(Are you aware of whether, when people come here, [palliative care] has already been discussed?) I don't always know that. Usually or often it is, but not always, or they haven't remembered it. But I don't know in advance, systematically, what they already know about it and what they don't." (Phase I physician)*

During the phase I trial, communication between the phase I unit and referring oncologists or GPs was notably limited. Although a system of notification letters is used by the phase I staff to keep the referring oncologist or GP up to date, the latter groups lamented the lack of useful information shared through these letters, often focusing on highly specific medical details and not including any information on palliative care. Referring oncologists noted that they often 'lose' a patient once they start participation in a phase I trial, even though they feel they might play a role in further follow-up, including palliative care. GPs specifically felt that a lack of communication resulted in them not being able to take up palliative care conversations with the patient, which they deemed to be a core task given their often longstanding relationship with the patient. Instead, they typically only see the patient once they are already dismissed from the trial, often much later than they had preferred to still play a meaningful role in their care. Patients indicated that once they started the oncological trajectory, their oncologist became their primary point of contact, leading them to less frequently visit the GP.

### Potential integration of palliative care in phase I clinical trials

**Perceived benefits of integration.** Patients typically indicated having no need for palliative care – at least at the stage where they were at. Informal caregivers were generally more open to the idea of standardized integration of palliative care in the phase I trajectories, noting that they themselves also had information and psychosocial support needs resulting from the patient's illness. They also expressed that some patients may not be sufficiently assertive to demand forms of palliative care if it is not actively offered to them. Healthcare providers identified several potential benefits of integrating palliative care into phase I trials, mainly centered around providing less aggressive care and broad forms of advance care planning. Normalization of palliative care could make them more accessible to patients and families – but also offering a clear framework for discussing palliative care to support healthcare providers who may be uncertain about how to broach the subject. Specifically, improved visibility of palliative care was noted as an important benefit, as many patients still remain unaware of the scope of palliative care and what it can offer them:

*"Yes, exactly, that we, yes, that visibility, that people know it exists and where they can possibly turn to and where it's located. I think that can indeed be improved." (Phase I physician)*

A standardized approach could prevent difficult conversations from being delayed until crisis moments, with improved expectation management being highlighted as an advantage, especially with the only small chances of effectiveness in mind:

*"We still offer a trial option, so there's still hope, but actually, that's only a small hope. And what if it goes wrong? Then we're suddenly back at the point where there are no options. So that can be anticipated." (Oncologist)*

Enhanced interdisciplinary communication was also viewed as a benefit, with a more integrated palliative care approach potentially ensuring that referring oncologists, GPs, and phase I teams remained aligned. GPs specifically felt that they had a more active role to play, yet were not given the opportunity to do so, as illustrated by the following quote:

*"I've experienced too often that patients die during the study. And where I then felt, darn, I would have wanted to be involved, to tell the patient: don't you see that what you're doing is pointless? Patients still die while receiving treatment." (GP)*

**Barriers to potential integration.**  Both patients and healthcare providers felt that bringing up palliative care while still receiving treatment could be perceived as giving conflicting messages, undermining patients' hope for therapeutic benefit that in turn influences their wellbeing. This conflict was at the heart of almost all interviews conducted:

*"I know that we are supposed to do advance care planning earlier, on paper, but just imagine sitting in front of a patient to whom you are still offering a treatment option, and at the same time, you have to say, 'we should also start thinking about your end of life.' That patient wouldn't understand what you're doing, because essentially, you're giving two contradictory messages." (Oncologist)*

Furthermore, phase I staff noted that integrating palliative care is complicated by the uncertain prognosis of phase I trial participants, many of whom are still in relatively good condition while in the trial, once again undermining the context necessary to start palliative care conversations. Conflicts were also intrinsic to the phase I staff, as illustrated by the following quote of a phase I physician who admits that despite the realization that trials have a small chance of success, the balance between this fact and their therapeutic role is a challenging one:

*"I think we're too focused, that we let it appear as if it's curative, too much still. I notice a tension among our doctors as well – that's what we struggle with as a team." (Phase I nurse)*

The current lack of communication and continuity in treatment between these providers also presents a barrier to this integration. Finally, healthcare providers noted hindrances by resource constraints (e.g., limited time and insufficient staff) and by the current healthcare system not financially incentivizing palliative care.

**Strategies for potential integration.**  Healthcare providers proposed several specific strategies to facilitate the integration of palliative care into phase I trials. These mainly included different suggestions for when palliative care conversations could be systematically offered to patients within the trial context. Some suggested scheduling a standardized palliative care conversation at trial initiation with a palliative care specialist, although many participants pointed out that the early stages of the trial are overloaded with information and that, for example, a month after initiation would be better suited. Others felt that it had to be offered at the end of the trial, at a minimum:

*"I think that upon ending phase I studies, regardless of whether it's a decision by the patient or exclusion due to medical study-related results, a conversation about palliative care and end-of-life decisions should be offered." (Palliative care specialist)*

Crucially, healthcare providers agreed that trajectories of patients, specifically with palliative care in mind, are too diverse in terms of prognosis and condition to strictly adhere to a structured approach – patients may not be open to such conversations. However, providers agreed that while palliative care should not be pushed onto the patients, it should nevertheless be more systematically brought up to note its potential benefits, with this currently being a missed opportunity due to patients not receiving the information they may, at some point, need:

*"I think you mainly have to offer [palliative care], that everyone should offer it a bit, and I think it's not bad if it's also offered somewhat in phase I, I think those people are very much in survival mode, and if that doesn't work, then they fall off, and then, well, if they don't know that safety net of palliative care exists, that's very unfortunate."* (Oncologist)

Other suggestions focused on practical tools. The development of a protocol outlining when and how palliative care should be introduced was suggested as a means to ensure consistency across providers, as expressed succinctly by the following GP:

*"Protocoling isn't just about rules. Through a protocol, you develop reflexes."* (GP)

Providing patients with accessible information about palliative care resources and incorporating quality-of-life monitoring tools, such as through standardized checklists and through utilizing the questionnaires that patients already have to complete for study purposes, were also recommended. Finally, electronic medical records were suggested as the most feasible way to improve information exchange between oncologists, GPs, and phase I staff, in language that is understandable for every healthcare provider and related to palliative care discussions.

## Discussion

This qualitative multiperspective interview study explored the current attention to quality of life in care for phase I oncological trial participants, the current role of palliative care, and the possibilities for better integrating palliative care in phase I oncological trials, from the perspectives of patients, informal caregivers, and various healthcare providers. Our findings indicate that palliative care is not systematically integrated into phase I trial settings in Belgium. Patients are neither proactively referred to palliative care specialists nor supported through generalist approaches, such as early advance care planning conversations. While trial participants generally reported a positive impact of participation on their quality of life, informal caregivers and healthcare providers questioned whether the support offered was sufficiently holistic, often noting that quality of life discussions remained limited to physical symptoms rather than general wellbeing. This reflects a paradox: although all stakeholders recognized the importance of quality of life, it often remained a reactive, and surface-level concern, rather than a fully integrated and proactive focus within trial care.

For patients with advanced cancer, enrolling in a phase I trial often served as a lifeline. Many participants described the trial as a source of hope and a means of staying in control – a chance to 'do something' when standard treatment options were exhausted. This hopeful mindset had emotional benefits: patients spoke of feeling optimistic simply by virtue of having access to a new treatment. However, this hope came with considerable burdens. Frequent hospital visits, stringent eligibility protocols, side effects, and the uncertainty of unproven therapies all placed strain on patients and their families. Patients were willing to endure these burdens as the price of maintaining hope. Within this context, the idea of involving palliative care was often met with resistance or indifference from patients, but also from healthcare providers, who feared that introducing palliative care discussions might undermine patients' hope. This mirrors broader findings that provider reluctance, rooted in concerns about destroying hope, continues to delay timely palliative care conversations [31–34]. Yet, previous research on hope in the phase I context has called for an inclusion of hope reframing within palliative care trajectories, shifting from supporting unrealistic hopes to nurturing realistic hopes [35,36]. Furthermore, palliative care interventions for patients participating in phase I studies have been shown to alleviate distress and improve quality of life for both patients and their caregivers, supporting the feasibility of integration of palliative care in the phase I context, as opposed to the feared incompatibility [9,10,37].

Patients' narrow view of palliative care as synonymous with terminal care is in line with international research and reflects a significant gap in communication [38]. Healthcare providers reinforced the view of patients. They rarely initiated

discussions about palliative care, consequently also not providing patients with information about the benefits it may provide. Although phase I staff noted that it is also their responsibility to discuss palliative care, in practice, the topic was frequently sidestepped or deferred, with providers assuming that someone else (such as a GP or the referring oncologist) would broach it at an appropriate time. Moreover, fragmentation among healthcare providers exacerbated this issue. Phase I trial specialists, referring oncologists, palliative care specialists, and GPs often operated in silos with poor coordination. There was no clear consensus on whose role it was to address advance care planning or introduce supportive care alongside the trial. This lack of communication not only reinforced misconceptions but likely contributed to the resistance patients displayed towards early palliative care engagement.

Informal caregivers and healthcare providers recognized important potential benefits of introducing palliative care into the phase I setting: mainly, improved holistic care and the normalization of advance care planning conversations before crises occur. Participants – especially health care providers – proposed several promising strategies. A common suggestion was to make palliative care discussions a routine part of the trial care pathway, while allowing the necessary flexibility. Rather than relying on individual providers to decide ad hoc when (or if) to mention palliative care, the idea is to standardize certain touchpoints for these conversations. These conversations could emphasize that services focused on quality of life are available in parallel with the trial, framing it as an additional layer of support rather than an alternative. Such an approach would ensure every patient is at least aware of palliative care resources, even if they choose not to use them immediately. Education and culture change were implicit in many of these strategies and has been suggested previously as an important part towards the integration of palliative care in phase I settings, by normalizing palliative care as a standard component of comprehensive cancer care (even in trials), hoping to gradually erode the stigma and fear surrounding it [36]. Referring oncologists also have a pivotal role in this cultural shift. Their earlier introduction of palliative care – ideally at the point of discussing referral to a phase I trial – can help set expectations and reduce stigma, while making it easier for phase I staff to continue the conversation in a new care context.

Our findings also highlight the missed opportunity to integrate quality of life questionnaires –patient-reported outcome measures (PROMs) – meaningfully into clinical practice. In contrast to phase III trials, where PROMs are often mandated, phase I staff members noted that PROMs are rarely utilized in phase I trials, in line with recent international literature [39–41]. Even when collected, PROM data was typically solely collected in the context of research purposes, leaving clinicians without systematic insight into patients' evolving quality of life needs. Strengthening PROM integration into clinical workflows could thus serve as an important stepping stone toward more holistic care approaches in phase I settings [42,43].

Strengths of this study include the triangulation of perspectives across multiple stakeholder groups, providing a nuanced understanding of both the barriers and opportunities for integration. A main limitation of our study is that the sample included only patients actively participating in phase I trials, potentially biasing findings towards more positive experiences and towards lower openness to palliative care than might be seen in patients who declined or discontinued participation. Additionally, selective recruitment to avoid patient burden may have resulted in an overrepresentation of individuals in relatively good health. Furthermore, our sample lacks diversity, with all participants being Dutch-speaking and predominantly male.

Future research should explore and evaluate models for the feasible and sustainable early integration of palliative care into phase I trial settings, based on our findings and previous literature [10]. Research has suggested future models of palliative care integration to rely more on generalist palliative care delivered by oncologists or clinical trial coordinators, who play a central role in the phase I trial process [44,45].

However, while previous palliative care interventions in the phase I setting have remained confined to that context [10], our findings suggest that this is a necessary but likely not sufficient step. It is also crucial to involve other healthcare providers who are part of the patient's care trajectory, particularly referring oncologists and general practitioners. These

professionals often have longstanding relationships with patients and may be best positioned to provide appropriate palliative care and initiate sensitive conversations. Furthermore, the systematic adoption of quality of life measures in the phase I trajectory may also be an opportunity for increased visibility of care for quality of life and palliative care, as highlighted by interviewed healthcare providers.

In conclusion, this qualitative study highlights that in current phase I oncological trials in Belgium, a palliative care approach is not systematically integrated into patient care trajectories. While the importance of quality of life was recognized in principle, it often remained an afterthought to the primary mission of drug dosing. Healthcare providers and informal caregivers acknowledged the potential benefits of integrating palliative care for holistic support and advance care planning, even as patients expressed little openness to the topic, associating it with terminal care. Important barriers exist, particularly around perceptions of incompatibility with active treatment and poor interdisciplinary communication, yet healthcare providers identified strategies for systematic yet flexible integration. These aim to ensure every patient is offered holistic support early and consistently, while respecting individual readiness and preserving hope. Overall, these findings underscore the need for structured, patient-centered approaches to support the complex needs of patients participating in phase I trials.

## Supporting information

**S1 File. S1 COREQ checklist.** COREQ checklist.
(PDF)

**S2 File. S2 Topic guides.** Topic guides.
(DOCX)

## Acknowledgments

We wish to thank Marthe Tulpin, Roselynn De Roeck, and Ruth Coussement for their help with conducting interviews, as well as the study coordinators of the Drug Research Unit Ghent and University Center for Clinical Research Antwerp for their help with data collection.

## Author contributions

**Conceptualization:** Frederick Daenen, Anne van Driessche, Anne-Lore Scherrens, Annelies Janssens, Sylvie Rottey, Lore Decoster, Peter A. J. Stevens, Koen Pardon, Aline De Vleminck, Kim Beernaert.

**Data curation:** Frederick Daenen, Anne van Driessche.

**Formal analysis:** Frederick Daenen, Anne van Driessche.

**Funding acquisition:** Anne-Lore Scherrens, Annelies Janssens, Sylvie Rottey, Lore Decoster, Kim Beernaert.

**Investigation:** Frederick Daenen, Anne van Driessche, Kim Beernaert.

**Methodology:** Frederick Daenen, Anne van Driessche, Anne-Lore Scherrens, Peter A. J. Stevens, Kim Beernaert.

**Project administration:** Frederick Daenen, Kim Beernaert.

**Resources:** Annelies Janssens, Sylvie Rottey, Lore Decoster.

**Supervision:** Kim Beernaert.

**Writing – original draft:** Frederick Daenen.

**Writing – review & editing:** Anne van Driessche, Anne-Lore Scherrens, Annelies Janssens, Sylvie Rottey, Lore Decoster, Peter A. J. Stevens, Koen Pardon, Aline De Vleminck, Kim Beernaert.

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
