## [Decision Letter · Decision Letter 0]

8 Sep 2025

PONE-D-25-38276Integration of palliative care into phase I oncology trials: A qualitative interview study with patients, informal caregivers, and healthcare providersPLOS ONE

Dear Dr. Daenen,

Thank you for submitting your manuscript to PLOS ONE. After careful consideration, we feel that it has merit but does not fully meet PLOS ONE’s publication criteria as it currently stands. Therefore, we invite you to submit a revised version of the manuscript that addresses the points raised during the review process.

Please note that we have only been able to secure a single reviewer to assess your manuscript. We are issuing a decision on your manuscript at this point to prevent further delays in the evaluation of your manuscript. Please be aware that the editor who handles your revised manuscript might find it necessary to invite additional reviewers to assess this work once the revised manuscript is submitted. However, we will aim to proceed on the basis of this single review if possible. 

If applicable, we recommend that you deposit your laboratory protocols in protocols.io to enhance the reproducibility of your results. Protocols.io assigns your protocol its own identifier (DOI) so that it can be cited independently in the future. For instructions see: https://journals.plos.org/plosone/s/submission-guidelines#loc-laboratory-protocols. Additionally, PLOS ONE offers an option for publishing peer-reviewed Lab Protocol articles, which describe protocols hosted on protocols.io. Read more information on sharing protocols at . Additionally, PLOS ONE offers an option for publishing peer-reviewed Lab Protocol articles, which describe protocols hosted on protocols.io. Read more information on sharing protocols at https://plos.org/protocols?utm_medium=editorial-email&utm_source=authorletters&utm_campaign=protocols..

We look forward to receiving your revised manuscript.

Kind regards,

Jennifer Tucker, PhD

Staff Editor

PLOS ONE

Journal Requirements:

3. In the online submission form, you indicated that as the data concern interview transcripts with participants, including patients, family caregivers, and healthcare professionals, and as they as such contain personal data, they cannot be shared publicly. Requests for pseudonymized versions of the data may be addressed to the first author. Every request will be evaluated on an individual basis and the ethics committee of the Vrije Universiteit Brussels will be contacted for approval before any sharing of participant-level data.

Reviewers' comments:

Reviewer's Responses to Questions

**Comments to the Author**

1. Is the manuscript technically sound, and do the data support the conclusions?

Reviewer #1: Yes

2. Has the statistical analysis been performed appropriately and rigorously? 

Reviewer #1: Yes

3. Have the authors made all data underlying the findings in their manuscript fully available?

Reviewer #1: Yes

4. Is the manuscript presented in an intelligible fashion and written in standard English?

Reviewer #1: Yes

5. Review Comments to the Author

Reviewer #1: I would like to thank the authors for their initiative. I t is important to assess the justification for using cancer patients with non-curative type with limited overall survival in Phase 1 trials. I would like to however, raise few points, and request the authors to address them, if possible,

1) Is there any method such as qualitative interview study? most qualitative studies use interviews as a data collection tool.

2) The authors can comment on the justification as available in literature to involve patients with such advanced stage in phase1 trials. They may briefly discuss the current scenario of PC in Belgium in general and the problems faced by the trial participants to indicate the current gaps.

3) The objectives 1 needs more clarity and objective two can be divided in two part.

4) Briefly mention how different perspectives will help will add to the strength of the study. This can come under the introduction.

5) Qualitative research does not opt for quantification and thus do not include inter-rater reliability. This is usually reached through triangulation.

6) Review the analysis for more contents/themes.

7) Some of the claims as reported under the result are difficult to understand unless supported by sufficient participant narratives. Also the probes made to the patients seem directive which may have prevented a deeper perspectives from surfacing.

8)The results indicate towards many related issues, such as bureaucratic complication, knowledge and attitude towards palliative care which can be either mentioned in the discussion or as future implications

I have mentioned these doubts in details in the manuscript as side notes.

Overall, the study has merits to add to the better knowledge of patient perspectives which are currently missing in healthcare many a times. Using them can make treatment and research patient-centric.

6. PLOS authors have the option to publish the peer review history of their article (what does this mean?). If published, this will include your full peer review and any attached files.). If published, this will include your full peer review and any attached files.

.

Reviewer #1: No

While revising your submission, please upload your figure files to the Preflight Analysis and Conversion Engine (PACE) digital diagnostic tool, https://pacev2.apexcovantage.com/. PACE helps ensure that figures meet PLOS requirements. To use PACE, you must first register as a user. Registration is free. Then, login and navigate to the UPLOAD tab, where you will find detailed instructions on how to use the tool. If you encounter any issues or have any questions when using PACE, please email PLOS at . PACE helps ensure that figures meet PLOS requirements. To use PACE, you must first register as a user. Registration is free. Then, login and navigate to the UPLOAD tab, where you will find detailed instructions on how to use the tool. If you encounter any issues or have any questions when using PACE, please email PLOS at figures@plos.org. Please note that Supporting Information files do not need this step.. Please note that Supporting Information files do not need this step.

---

## [Author Response · Author response to Decision Letter 1]

10 Nov 2025

All reviewer and editor comments have been addressed in the attached reponse letter to reviewers.

---

## [Decision Letter · Decision Letter 1]

22 Mar 2026

PONE-D-25-38276R1Integration of palliative care into phase I oncology trials: A qualitative interview study with patients, informal caregivers, and healthcare providersPLOS One

Dear Dr. Daenen,

Thank you for submitting your manuscript to PLOS ONE. After careful consideration, we feel that it has merit but does not fully meet PLOS ONE’s publication criteria as it currently stands. Therefore, we invite you to submit a revised version of the manuscript that addresses the points raised during the review process.

If applicable, we recommend that you deposit your laboratory protocols in protocols.io to enhance the reproducibility of your results. Protocols.io assigns your protocol its own identifier (DOI) so that it can be cited independently in the future. For instructions see: https://journals.plos.org/plosone/s/submission-guidelines#loc-laboratory-protocols. Additionally, PLOS ONE offers an option for publishing peer-reviewed Lab Protocol articles, which describe protocols hosted on protocols.io. Read more information on sharing protocols at . Additionally, PLOS ONE offers an option for publishing peer-reviewed Lab Protocol articles, which describe protocols hosted on protocols.io. Read more information on sharing protocols at https://plos.org/protocols?utm_medium=editorial-email&utm_source=authorletters&utm_campaign=protocols..

We look forward to receiving your revised manuscript.

Kind regards,

JONATHAN BAYUO, PhD

Academic Editor

PLOS One

**Journal Requirements:**

**Additional Editor Comments:**

Thanks to the authors for addressing the comments previously raised. Some minor concerns require further attention. Please see the comments below:

1. Kindly highlight the exact qualitative design that was employed. This should be made explicit under the study design section with appropriate references.

2. Ethical approval and considerations have been reported under "population and setting". Please provide a distinct section for this.

3. There is no section on trustworthiness or methodological rigour. This is central to qualitative research and must be included. Kindly create a specific section for this under the methods section.

Reviewers' comments:

Reviewer's Responses to Questions

**Comments to the Author**

1. If the authors have adequately addressed your comments raised in a previous round of review and you feel that this manuscript is now acceptable for publication, you may indicate that here to bypass the “Comments to the Author” section, enter your conflict of interest statement in the “Confidential to Editor” section, and submit your "Accept" recommendation.

Reviewer #1: All comments have been addressed

2. Is the manuscript technically sound, and do the data support the conclusions?

Reviewer #1: Yes

3. Has the statistical analysis been performed appropriately and rigorously? 

Reviewer #1: Yes

4. Have the authors made all data underlying the findings in their manuscript fully available?

Reviewer #1: Yes

5. Is the manuscript presented in an intelligible fashion and written in standard English?

Reviewer #1: Yes

6. Review Comments to the Author

Reviewer #1: (No Response)

7. PLOS authors have the option to publish the peer review history of their article (what does this mean?). If published, this will include your full peer review and any attached files.). If published, this will include your full peer review and any attached files.

.

Reviewer #1: No

---

## [Author Response · Author response to Decision Letter 2]

30 Mar 2026

Responses to the editor/reviewer comments have been included in the accompanying 'Response to Reviewers' document.

---

## [Editor Report · Decision Letter 2]

31 Mar 2026

Integration of palliative care into phase I oncology trials: A qualitative interview study with patients, informal caregivers, and healthcare providers

PONE-D-25-38276R2

Dear Dr. Daenen,

We’re pleased to inform you that your manuscript has been judged scientifically suitable for publication and will be formally accepted for publication once it meets all outstanding technical requirements.

An invoice will be generated when your article is formally accepted. Please note, if your institution has a publishing partnership with PLOS and your article meets the relevant criteria, all or part of your publication costs will be covered. Please make sure your user information is up-to-date by logging into Editorial Manager at Editorial Manager® and clicking the ‘Update My Information' link at the top of the page. For questions related to billing, please contact  and clicking the ‘Update My Information' link at the top of the page. For questions related to billing, please contact billing support..

Kind regards,

JONATHAN BAYUO, PhD

Academic Editor

PLOS One

Additional Editor Comments (optional):

Thanks to the authors for thoughtfully addressing the comments raised.
---

## [Editor Report · Acceptance letter]

PONE-D-25-38276R2

PLOS One

Dear Dr. Daenen,

I'm pleased to inform you that your manuscript has been deemed suitable for publication in PLOS One. Congratulations! Your manuscript is now being handed over to our production team.

Kind regards,

on behalf of

Dr. JONATHAN BAYUO

Academic Editor

PLOS One